# Advances in Immunotherapy for Hepatitis B

**DOI:** 10.3390/pathogens11101116

**Published:** 2022-09-28

**Authors:** Dongyao Wang, Binqing Fu, Haiming Wei

**Affiliations:** 1Department of Hematology, the First Affiliated Hospital of USTC, Division of Life Sciences and Medicine, University of Science and Technology of China, Hefei 230027, China; 2Blood and Cell Therapy Institute, Division of Life Sciences and Medicine, University of Science and Technology of China, Hefei 230001, China; 3Institute of Immunology and the CAS Key Laboratory of Innate Immunity and Chronic Disease, School of Basic Medicine and Medical Center, University of Science and Technology of China, Hefei 230001, China; 4Anhui Provincial Key Laboratory of Blood Research and Applications, Hefei 230001, China; 5Anhui Province Key Laboratory of Immunology in Chronic Diseases, Bengbu Medical College, Bengbu 233030, China

**Keywords:** HBV, Immunotherapy, immune evasion

## Abstract

Hepatitis B virus (HBV) is a hepatotropic virus with the potential to cause chronic infection, and it is one of the common causes of liver disease worldwide. Chronic HBV infection leads to liver cirrhosis and, ultimately, hepatocellular carcinoma (HCC). The persistence of covalently closed circular DNA (cccDNA) and the impaired immune response in patients with chronic hepatitis B (CHB) has been studied over the past few decades. Despite advances in the etiology of HBV and the development of potent virus-suppressing regimens, a cure for HBV has not been found. Both the innate and adaptive branches of immunity contribute to viral eradication. However, immune exhaustion and evasion have been demonstrated during CHB infection, although our understanding of the mechanism is still evolving. Recently, the successful use of an antiviral drug for hepatitis C has greatly encouraged the search for a cure for hepatitis B, which likely requires an approach focused on improving the antiviral immune response. In this review, we discuss our current knowledge of the immunopathogenic mechanisms and immunobiology of HBV infection. In addition, we touch upon why the existing therapeutic approaches may not achieve the goal of a functional cure. We also propose how combinations of new drugs, and especially novel immunotherapies, contribute to HBV clearance.

## 1. Introduction

Hepatitis B virus (HBV) is a prototypical member of the Hepadnaviridae family. HBV is a hepatotropic virus that generates covalently closed circular (ccc) DNA, a plasmid-like episome, in the nucleus of host cells [1,2,3]. According to the estimated number of the world’s population with serological evidence of current or past HBV infection, around 2 billion people may have been infected with HBV at some point in their life [4,5,6]. Many of these infections are acquired in infancy or early childhood and could lead to chronic hepatitis B (CHB), which is highly prevalent in some parts of Africa and Asia [6,7]. There are more than 250 million individuals infected with CHB worldwide (at the time of writing) [7]. Furthermore, ~700,000 deaths per year are caused by complications of persistent HBV infection, liver cirrhosis, as well as hepatocellular carcinoma (HCC) [2,8,9,10,11,12]. In China, the burden of HBV is considerable [13,14]. It was reported that, in 21–49-year-old men, the seroprevalence was ~6% [15]. For younger individuals (aged between 1 and 29 years old), the reported incidence was ~2.6% [16]. This lower figure can be attributed to the success of vaccination policy, which caused the seroprevalence of hepatitis B surface antigen (HBsAg) in younger people to decline rapidly. 

In patients with CHB, a high HBV load, and serum hepatitis B e antigen (HBeAg) and HBsAg levels may play a key role in the impaired antiviral immune response [17]. However, the mechanism has not been fully explored for various reasons. Preventive treatments, including a prophylactic vaccine, have a significant effect on HBV control [18]. However, the vaccine does not appear to be beneficial for people with existing CHB. In addition, current hepatitis B therapies are limited to immunoregulatory drugs, including IFN-α, or several direct-acting antivirals (DAAs) such as tenofovir and entecavir (ETV) [10]. Moreover, the drugs rarely achieve HBV clearance from the liver, meaning that the majority of patients need lifelong treatment [19]. The target of any new treatment regimen is to increase the possibility of a functional cure [20]. Current therapies have improved the quality of life and the survival of patients with CHB, and reduced the incidence of HCC, cirrhosis, and other complications through the suppression of HBV replication and/or by reducing hepatic necroinflammation. However, the ultimate goal of a functional cure is not frequently achieved.

In this review, we have summarized the key milestones of HBV research that has been performed over the last 30 years and focused on recent findings relating to advances in the etiology of HBV and immunologic assumptions. As the important challenge of achieving an HBV functional cure is likely to be overcome by improving the HBV-specific immune response, we have also reviewed the current strategies aimed at restoring the function of HBV-specific immune cells. 

## 2. The Etiology for Hepatitis B

In China, prior to vaccination, hepatitis B was typically spread through vertical transmission. Nearly 20% of HBsAg-positive families contain at least two HBsAg carriers [13,21]. In Asia, since hepatitis B occurs at an early age, CHB and viral persistence seem more frequent, complicating the selection of effective treatment options. The HBV genotypes vary across different geographical regions. In Europe and the United States, genotypes A and D are the most frequently occurring HBV genotypes, while genotypes B and C are predominately found in China [22]. Many important factors, such as the genotype, the age of the infected individual, as well as the stage of the disease, could influence the immune response to therapy.

The entry of HBV into host cells is a complex process. Parenchymal liver cells are susceptible to infection upon HBV entering the circulatory system. Features of human liver microcirculation, including slow blood flow, have been demonstrated to increase the possibility of HBV interacting with the sodium taurocholate co-transporting polypeptide (NTCP), which is expressed on the surface of hepatocytes. This interaction is thought to initiate viral entry into host cells and its subsequent replication [23,24,25]. Additionally, platelets are always recruited to the liver microcirculation after viral infection. Their activation correlates with severely reduced microcirculation, and delayed viral elimination. Lack of platelet-produced serotonin contributes to accelerating viral clearance in the liver. 

HBV then spreads rapidly throughout the liver. Studies performed in chimpanzees verified that, during the initial spreading stage, HBV can easily escape recognition by the innate immune system. This immune escape is probably due to the unique replication strategy of HBV, which involves the cccDNA molecule [2]. Furthermore, cccDNA has been confirmed to be the source of the circulating antigens HBsAg and HBeAg in human peripheral blood. This replicative feature enables HBV to produce a viral load of over a billion particles per milliliter [17]. Moreover, studies of human liver biopsies did not find significant innate immune responses in the early phase of CHB. In addition, it was reported that HBV may have evolved a hidden strategy to evade recognition by the innate immune system and rapidly infect and replicate in the hepatocytes [26,27]. 

HBV DNA is considered to be a major biomarker of viral replication and has been regarded as an important endpoint of clinical trials using nucleoside analog therapy [28,29]. Hepatitis B virions have an envelope containing three viral gene products, including HBsAg determinant [2,30]. The HBV envelope has been found to enclose an inner nucleocapsid particle, which in most virions, is composed of 120 core protein (i.e., HBcAg) homodimers. Furthermore, it has been shown that the nucleocapsid particles include a copy of a partially double-stranded relaxed circular DNA (rcDNA) genome (Figure 1) [31,32]. Approximately 10% of the nucleocapsid particles are thought to contain double-stranded linear DNA (dslDNA) in place of an rcDNA genome [2,33,34]. After HBV entry, the nucleocapsids (with their rcDNA) are transported into the nucleus, where the host enzymes participate in the repair of the viral genome and its conversion into the cccDNA [35,36,37].

## 3. Mechanisms of HBV Immune Evasion

HBV can avoid elimination by the immune system via a process called immune evasion, which is a major concern in CHB. There are various mechanisms of HBV immune evasion, including reduced TNF-α production by T cells and Kupffer cells, impaired IFN-α production by plasmacytoid DC (pDC) cells, low induction of interferon-stimulated genes (ISGs), inhibition of TLR signaling [38,39,40,41,42,43,44,45,46], and so on. Although immune evasion may implicate both the innate and adaptive branches of immunity, the exact mechanisms remain unknown. 

### 3.1. Evasion of the Adaptive Immune Response

Over the last 30 years, numerous studies in humans and animal models have demonstrated that the outcome of HBV infection is strongly determined by the dynamics and the effector functions of the HBV-specific adaptive immune response. Adaptive immunity is a complex branch of the human immune system, and the main force against HBV [2]. During HBV infection, both the number and fitness of HBV-specific T and B lymphocytes increase significantly [47,48,49]. The indispensable role played by HBV-specific CD8^+^ T cells in the clearance of HBV has been well recognized [38,48,50,51,52,53]. Moreover, CD4^+^ T cells are required to promote the activation and function of these CD8^+^ T cells. However, these adaptive immune responses are typically functionally impaired in patients with CHB [2,47,53,54,55,56,57]. It has been reported that HBV-specific CD8^+^ T cells in both the peripheral blood and liver microenvironment of patients with CHB always exhibit an exhausted phenotype [58,59]. Furthermore, in patients with CHB, suppressive mechanisms, such as regulatory T cells (Tregs), and the increased expression of co-inhibitory receptors such as cytotoxic T-lymphocyte antigen 4 (CTLA-4), T cell immunoglobulin and mucin domain 3 (Tim-3), and programmed cell death protein 1 (PD-1) on CD8^+^ T cells dampen the antiviral response [10,53,60,61,62]. 

Although the mechanisms that regulate the above co-inhibitory factors are still to be elucidated, hepatocytes and other hepatic cell populations might contribute to the impaired function and exhaustion of HBV-specific CD8^+^ T cells. It was reported that the hepatocytes of patients with CHB did not express certain classical costimulatory molecules (e.g., CD80 and CD86), meaning that infected hepatocytes may not be able to transfer the second signal required for CD8^+^ T cell activation [63,64]. In addition, Benechet et al. found that naïve HBV-specific CD8^+^ T cells primed by infected hepatocytes could not differentiate into interferon (IFN)-γ-expressing effector T cells in an HBV transgenic mouse model [65]. Tregs and macrophages may also contribute to the immunosuppressive environment through the production of immunomodulatory cytokines such as interleukin (IL)-10 and tumor growth factor (TGF)-β1 [39,66,67]. Furthermore, myeloid-derived suppressor cells (MDSC) in the liver microenvironment were shown to suppress T cell signaling partially through the production of arginase, which could degrade arginine and significantly inhibit T cell effector function [68,69,70].

Antibodies secreted by B cells recognize antigens by directly binding to the components of the pathogen or interacting with the proteins expressed on the surface of hepatocytes [48]. However, only antibodies targeting the HBV envelope can prevent the spread of HBV infection [57,71]. In addition, the presence of HBsAg in the serum causes B cell exhaustion and impedes the maturation of HBsAg-specific B cells, which might be associated with the high expression of PD-1 [72]. Indeed, PD-1 blockade or HBsAg clearance were shown to restore the antibody-producing ability of HBsAg-specific B cells [73,74]. Moreover, persistent IFN-α therapy was able to induce large numbers of CD24^+^CD38^hi^ regulatory B cells (Bregs) and promote an immunosuppressive response, which resulted in the downregulation of CD8^+^ T cell and natural killer (NK) cell effector functions (Figure 2). In addition, patients with fewer Bregs exhibited improved therapeutic effects [53]. Some other types of cells also pay a role during HBV infection such as the liver-resident macrophages, the Kupffer cells [75]. Kupffer cells are regarded as one of the predominant populations in the liver and they secrete immunomodulatory cytokines, for example, TGF-β1 and IL-10 [38]. Additionally, Kupffer cells in the liver highly expressed PD-L1 or PD-L2 during CHB infection, thus suppressing antiviral immune responses and leading to immune tolerance [38,75]. However, Kupffer cells could also present antigens to CD8^+^ T cell and induce their activation. Moreover, Kupffer cells have been found to recruit monocytes via chemokines in the liver [75]. Tacke et al. reported that the pathogenic macrophage subsets were a potential target for treating liver disease in mouse models [75]. Furthermore, there are enriched NKT cells in the human liver, which also play important roles in CHB and could modulate both innate and adaptive immunity [40,76]. The MHC-like molecule CD1d is important for NKT cells to recognize the lipid-based antigen [40,77,78]. However, until now, the characterization of NKT cells in the liver of CHB patients has still been poorly verified. It has been reported that, in the HBV-infected liver, the proportion of NKT cells were obviously decreased and had lost a-galactosylceramide (a-GalCer)-induced IFN-γ production, which may contribute to immune evasion. Importantly, when PBMCs were stimulated with α-GalCer plus IL-2 and IL-15, the ratio and the IFN-γ production of NKT cell were restored [76,77], which indicated that protective immunity might be partially recovered in patients with CHB. Taken together, these findings imply that the mechanisms that impair T and B cell responses during CHB likely contribute to the evasion of the adaptive immune response.

### 3.2. Evasion of the Innate Immune Response 

Although T cell dysfunction may be the most fascinating immune change leading to HBV persistence, the interactions between HBV and different types of innate immune cells should not be ignored. However, the relationship between the virus and innate immune response remains highly controversial, and the mechanisms of innate immune escape have not been fully understood [17].

HBV can disrupt antiviral responses within infected cells and evade the innate immune response. Pattern recognition receptor (PRR) distribution has been identified as an important factor for identifying drugs targeting the innate immune system [79,80]. During chronic HBV infection, interactions between HBV antigens, PRRs, as well as various innate immune cell types, have been reported [17]. For instance, dendritic cells (DCs) promote the adaptive immune response through antigen presentation and the production of several cytokines, including IL-12 [81]. HBV may suppress DCs by downregulating the expression of co-stimulatory molecules such as CD80 and CD86, and by inducing the high expression of PD-L1 [82,83,84]. Additionally, it has been reported that the proportion of plasmacytoid DCs (pDCs) was markedly decreased in patients with CHB [85,86,87]. It was also reported that monocytes inhibit the production of IL-12 by DCs when exposed to HBsAg [88]. Therefore, targeting innate immunity may be a potential novel approach to developing a functional cure for HBV infection in the future.

## 4. Progress in Hepatitis-B-Specific Immunotherapy

Existing treatment regimens have achieved remarkable cure rates for patients infected with the hepatitis C virus (HCV). However, the current regimens for treating HBV remain suboptimal. Current therapeutic approaches include nucleoside analogues (NA) and nucleotide drugs (NUCs), which both efficiently inhibit HBV replication. Lamivudine, the first nucleoside reverse transcriptase inhibitor (NRTI), obtained Food and Drug Administration (FDA) approval in 1998. Since then, other NRTIs such as adefovir and telbivudine have been developed but these are not used as first-line therapies due to drug-associated resistance. Currently, entecavir (ETV), tenofovir alafenamide (TAF), and tenofovir disoproxil fumarate (TDF) are used as the first-line oral drugs against HBV infection [89,90]. These agents can optimally lower HBV DNA levels in the serum of patients and reduce liver failure. However, current antiviral agents have minimal impact directly on cccDNA in primary human hepatocytes [80]. With the persistence of long-lived cccDNA, the potential for relapse of HBV exists, even after the clearance of viremia [91]. In addition, integrated viral DNA may survive immune clearance and the potential for relapse also exists in patients with resolved HBV infection. To date, the study of HBV cccDNA is still hampered by the lack of an appropriate model [92,93]. A deeper understanding of cccDNA might provide new perspectives to find a functional cure. Additionally, the loss of HBeAg or HBsAg with prolonged therapy occurs in very few patients [17,27]. For HBeAg-positive patients, oral antiviral drugs are regarded as the most common treatment strategy because of their effectiveness and ability to provide sustained viral suppression. The decision to treat HBeAg-positive CHB patients with one of the NUCs (such as lamivudine, entecavir, or tenofovir) should be individualized [94,95]. Additionally, the number of HBeAg-negative CHB patients is increasing and these patients have become the majority in terms of the form of chronic HBV, especially in Middle Eastern and north African countries. Indeed, few patients who are HBeAg-negative would achieve the loss of HBsAg, and a large quantity of these patients may experience HBV recurrence after discontinuation of therapy. Therefore, most guidelines suggest lifelong treatment, with the goal of achieving high rates of viral suppression [96]. Marcellin et al. carried out a study of HBeAg-negative patients with TDF treatment for up to 10 years and demonstrated that TDF therapy resulted in persistent maintenance of viral suppression and was well tolerated [97]. Furthermore, through a study from 17 countries, Maria Buti found that more than 90% of patients who were HBeAg-negative and receiving TDF had an HBV DNA of less than 29 IU/mL after treatment for 48 weeks [98].

IFN-α has antiviral properties and can regulate immune function. To date, IFN-α has been regarded as the first-choice therapy for treating CHB [10]. A multi-center study reported that, after 48 weeks of treatment with a combination of PEG-IFN-α plus TDF, a 9.1% HBsAg loss was observed at week 72 post treatment initiation [99,100,101]. In addition, Fu et al. reported that, after PEG-IFN-α-2b treatment, approximately 30% of patients with CHB underwent HBeAg seroconversion by week 72 [53,54]. However, there are some limitations of therapeutic IFN-α administration. For example, IFN-α therapy may have some side effects and contraindications, especially in patients with advanced liver disease [102]. To achieve a functional cure for HBV, a properly orchestrated activation of anti-HBV immunity is required. As patients with CHB have low numbers of HBV-specific CD8^+^ T cells, which are frequently exhausted, existing immunotherapeutic approaches designed to promote antiviral immunity may not be adequate [103,104,105]. After analyzing the characteristics of innate and adaptive immune response during HBV infection, some promising immuno-dependent therapeutic strategies to achieve a functional cure for CHB were proposed. These included IL-2, checkpoint inhibitors (e.g., anti-PD-1 and anti-CTLA-4), and therapeutic vaccines. We demonstrated that non-responder (NR) patients who failed to respond to PEG-IFN-α treatment, benefited from a sequential low dose of IL-2 (1 × 10^6^ IU) therapy, which caused a decrease in the frequency of PD-1^+^ CD8^+^ T cells and Tregs [9]. Furthermore, we found sequential IL-2 therapy significantly restored the frequency of HBV-specific CD8^+^ T cells and the HBsAg-specific effector function of CD8^+^ T cells. Importantly, we found that, in the majority of NR patients, HBeAg levels were markedly decreased after sequential IL-2 therapy [10] (Figure 3).

Checkpoint inhibitors and therapeutic vaccination have also been proposed to restore the antiviral immune response of patients with CHB [106]. However, so far, treatments with checkpoint inhibitors have only been applied usefully in some solid malignancies including melanoma and renal cell carcinoma. In vitro studies have shown that anti-PD1/PDL-1 blockade could partially restore the function of exhausted HBV-specific CD8^+^ T cells [62,107,108]. Gane et al. found that nivolumab (a PD-1 inhibitor) with or without GS-4774 (a therapeutic vaccine) was well-tolerated and would contribute to an HBsAg decrease in virally suppressed HBeAg-negative patients in a phase Ib study [109]. In addition, trials of several vaccine candidates have been carried out in patients with CHB [18,110,111]. Boni et al. reported that the administration of tenofovir plus GS-4774 therapy was well tolerated and could improve HBV-specific T cell responses in CHB patients. In addition, the production of TNF-α, IFN-γ, as well as IL-2, obviously increased. Furthermore, data have suggested that combination treatments including vaccines may be regarded as sequential administration that is able to increase the antiviral immune response in the future [112].

The role of the innate immune system should not be ignored in the process of HBV eradication. However, in the context of HBV infection, the innate immune response is often poorly activated due to immune evasion. RIG-I or Toll-like receptor (TLR) agonists, such as TLR-7 and TLR-8 agonists, have been used to induce the activation of innate immunity. GS-9620, a TLR-7 agonist, has been found to induce the production of IFN-α, especially by pDCs. In addition, the treatment of RO7020531 triggered obvious immune activation in patients with CHB [113,114,115]. Moreover, recently developed TLR-8 agonists may contribute to the activation of PRRs present in the liver, and GS-9688 has been shown to promote the production of IL-12 and IL-18 from monocytes or DCs [116,117,118]. Furthermore, as cytokines such as IL-12 also contribute to NK cell activation, which have been demonstrated to kill both HBV-infected hepatocytes and HBV-specific CD8^+^ T cells, it is necessary to comprehensively evaluate the function of activated innate immunity in the process of HBV eradication. Furthermore, there are a large quantity of new therapeutic drugs for patients with CHB under investigation. GLS4 is a core protein allosteric modulator. A total of 20 weeks of treatment with GLS4 resulted in reduced DNA levels (1.48–6.09 log decrease) [20]. Several capsid assembly modulators have been under development for CHB therapy. For example, ABI-H0731, was found to cause a significant decrease in HBV DNA levels at 12 weeks, when combined with entecavir [119]. In addition, the administration of RO7049389 not only reduced HBV DNA levels, but also decreased HBsAg, as well as HBeAg levels in the serum [20,120]. Additionally, the effects of siRNAs in clinical trials also appear encouraging. Treatment with JNJ3989 achieved a 1.3–3.8 log decrease in HBsAg levels [20]. Furthermore, several other new therapies that have been investigated have been reported as safe and well tolerated in healthy volunteers, such as GSK3389404 [19] (Table 1).

Furthermore, immunological approaches against HBV infection, which involve the use of T cells engineered with a classical T cell receptor (TCR) specific for human leukocyte antigens (HLA)-class I restricted HBV epitopes or a chimeric antigen receptor (CAR), have shown some promise [106]. Despite the encouraging preliminary results of such T cell therapies, they are associated with a risk of inducing fatal hepatic inflammation. Thus, the adoptive transfer of engineered T cells and the manufacturing techniques used must be evaluated with more caution. In addition, more robust experimental and clinical trial data are needed. Platelets play important roles in inflammatory and immune-mediated disorders. Aiolfi et al. reported that platelets contributed to the pathogenesis of HBV-related liver disease by their ability to promote the homing of effector CD8^+^ T cells in the liver, expression of pro-angiogenic mediators (such as VEGF and TGF-β1) and the production of pro-inflammatory cytokines (such as IFN-γ) [130]. Aspirin is a widely used anti-platelet drug. Notably, the suppression of platelet activation using aspirin would significantly reduce the number of HBV-specific CD8^+^ T cells and the recruitment of inflammatory cells in the liver, which contributes to alleviating liver injury and the likelihood of HCC [130,131]. Therefore, anti-platelet therapy might be another promising approach for the treatment of patients with CHB. Collectively, to achieve the goal of developing a functional cure, more knowledge derived from the accumulation of experiment and clinical trials is needed.

## 5. Conclusions and Perspectives

Based on the findings presented, it is clear that the future of HBV treatment requires the direct suppression of cccDNA replication. However, achieving the ultimate goal of finding a functional cure for CHB will be challenging. Although the molecular biology of HBV is becoming gradually understood and novel DAAs are being developed, it is still unclear whether these agents are safe and would be able to provide a long-term functional cure. Immunotherapy is receiving increasing attention from scientists and clinicians in many fields of research. Additionally, the investigation of curative strategies for patients with CHB will benefit greatly from the knowledge of immunological features and mechanisms that govern HBV pathogenesis and immunobiology. Despite ongoing challenges in the quest for HBV eradication, there remains much promise and optimism on the way to achieving the goal of an HBV functional cure.

## Figures and Tables

**Figure 1 pathogens-11-01116-f001:**
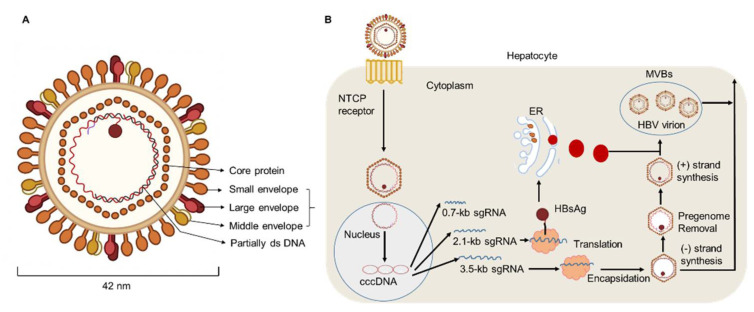
**HBV particle and life cycle.** (**A**) Hepatitis B virions are about 42 nm in diameter. The envelope of HBV virion contains three forms of HBsAg: large (L), middle (M), and small (S) envelope proteins. The capsid encapsidates a partially double stranded (ds) DNA. The HBV envelope has an inner nucleocapsid particle that always consists of 120 core protein. (**B**) Firstly, HBV attaches to the host cell membrane through its envelope proteins and the sodium taurocholate co-transporting polypeptide (NTCP). Next, the viral genome reaches the cytoplasm of hepatocytes and enters the nucleus, where host enzymes will repair the genome into the covalently closed circular DNA (cccDNA). In addition, transcription and nuclear export of mRNA to the hepatocellular cytoplasm for translation are observed. HBsAg are produced via the endoplasmic reticulum (ER)-Golgi complex and then assembled in the cytoplasm, while HBV virions are formed by budding from multivesicular bodies (MVBs). The new virions will exit the host and infect new hepatocytes.

**Figure 2 pathogens-11-01116-f002:**
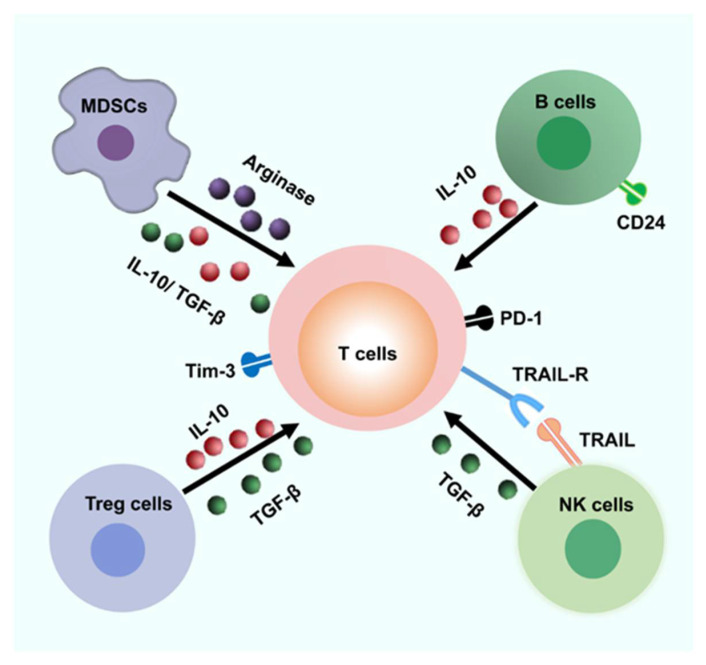
**Possible inhibitory mechanisms result in immune escape.** The liver microenvironment is enriched with various cells that can inhibit the T cell responses. Abnormal PD-1, Tim-3, and other negative signaling pathways probably result in the T cell exhaustion. MDSCs and Tregs could be a great source of arginase and suppress T cell responses through arginase and secreting several immune-modulatory cytokines such as TGF-β1 and IL-10. In addition, CD24^+^ Breg cells may suppress T cell function by IL-10. Furthermore, HBV-specific CD8^+^ T cells could be lysed by activated NK cells via a contact-dependent manner (for example, TRAIL/TRAIL-R). MDSC, Myeloid-derived suppressor cells; TRAIL, tumor necrosis factor-related apoptosis-inducing ligand.

**Figure 3 pathogens-11-01116-f003:**
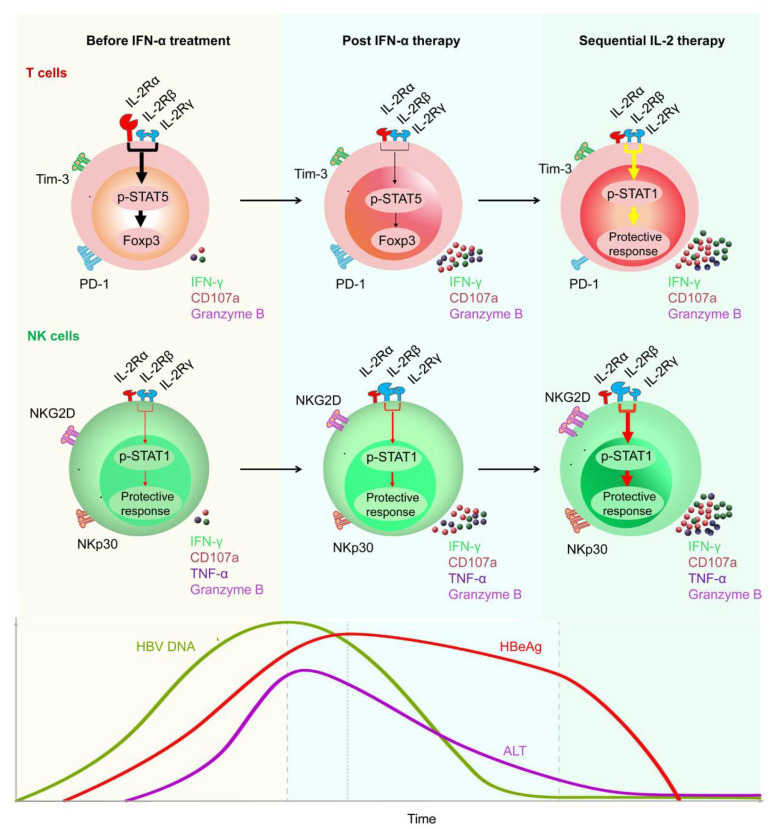
Restoration of HBV-specific CD8^+^ T cell and NK cell responses by sequential IL-2 treatment in non-responder patients after IFN-α therapy. IL-2Rα was high expressed, and NKp30 was low expressed on T cells and NK cells, respectively, of non-responder (NR) patients, in whom IFN-α therapy had failed. Those NR patients were treated with low-dose IL-2 for 24 weeks. A decrease in IL-2Rα expression on their CD4^+^ T cells was verified, suggesting that IFN-α therapy may provide a rationale for sequential IL-2 treatment without increasing regulatory T cells (Tregs). In addition, non-responders experienced a decrease in the numbers of PD-1 expression. Furthermore, sequential IL-2 administration restored effective immune function, involving STAT1 activation in both T cells and NK cells. Moreover, IL-2 therapy increased the function of HBV-specific T cells and NK cells, which translated into improved clinical outcomes, including HBeAg seroconversion, among the non-responder CHB patients.

**Table 1 pathogens-11-01116-t001:** Select new therapeutic strategies for patients with CHB under development.

Drug Names	Mechanism of Function	Effects	References
GLS4	Core binding	Data of 20 weeks demonstrated DNA level log decrease of 1.48–5.58 after administration (twice daily, BID)	[20]
ABI-H0731	Core binding	Data showed mean maximum NA level log reduction from baseline were 1.7, 2.1, and 2.8 in the 100, 200, and 300 mg dose group, respectively	[20,119]
RO7049389	Core binding	Median DNA level declines of 2.7 (200 mg, BID) and 3.2 (400 mg, BID) demonstrated at 28 days	[20,121]
REP 2165	HBsAg binding	Obviously higher percentages of CHB patients in REP 2165 group had reduction in HBsAg to below 1 IU/mL and HBsAg seroconversion during the first 24 weeks of TDF and PEG-IFN-α treatment	[20,122,123]
TG-1050	Transgene	HBV specific T cell responses were induced. Data at day 197 showed mean 0.45 log reduction in HBsAg levels	[124]
RO7020531	TLR7 agonist	Safety and tolerability in healthy Chinese donors with a 150 mg q.o.d.	[125]
GS-9688	TLR8	The antiviral efficacy of 3 mg/kg (weekly) was confirmed in a woodchuck study	[126]
JNJ3989	mRNA degradation	Data showed HBsAg level log reduction of 1.3–3.8	[127]
CRV431	Blocks NTCP	Data showed a significantly decreased liver HBV DNA levels with the treatment (50 mg/kg/day) for 16 days	[128]
GSK3389404	mRNA degradation	Data showed safe and target engagement, with dose-dependent reductions in HBsAg	[129]

HBsAg, hepatitis B surface antigen; HBV, hepatitis B virus; BID, twice daily; anti-HBs, anti-hepatitis B surface protein; TLR, Toll-like receptor; IU, infectious units; NTCP, sodium–taurocholate cotransporting polypeptide.

## Data Availability

All data referred to this study are available in the manuscript.

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
