# Peer review of "Advances in Immunotherapy for Hepatitis B"

_pathogens, 2022, doi:10.3390/pathogens11101116_

Round 1

Reviewer 2 Report

The manuscript by Wang et al. focuses on new aspects of etiology and treatment strategy of hepatitis B virus (HBV). Other reviews (some cited by the authors), recently discussed about immunobiology, pathogenesis and immunotherapy of HBV infection.

The present review is synthetic and the subject is treated in a simple manner, making it easy to read also to physicians different from infectivologists or hepatologists.

 Major comments

-          The author should improve the presentation of the new progress of immunotherapy. There are some interesting papers that should be added to the references and in the text [ i.e. Gane E, et al. Anti-PD-1 blockade with nivolumab with and without therapeutic vaccination for virally suppressed chronic hepatitis B: a pilot study. J Hepatol. 2019;  Boni C, et al Combined GS-4774 and Tenofovir Therapy Can Improve HBV-Specific T-Cell Responses in Patients With Chronic Hepatitis, Gatroenterology 2019; ……..….]

-          The review should discuss immunotherapic approach in HBeAg positive from antiHBe positive chronic hepatitis B.

-          The role of platelets and antiplatelet drugs in treatment of chronic hepatitis B should be added

Minor comments

-          The authors have discussed in details the results of their own study; more balance in the presentation of the results of different studies is required.

Reviewer 3 Report

The manuscript “The progress of etiology and immunotherapy for hepatitis B” is giving a brief review of current immunotherapies for HBV. It is not clear what is meant by “the progress of etiology  for hepatitis B” . The so named paragraph is not really bringing novel information about pathogenesis of the infection but rather some known facts, despite current references. I suggest rewriting this part and omitting it from the title. Greater elaboration of the data given in table 1 would improve the quality of the review. There are language problems throughout the text.

Round 2

Reviewer 2 Report

I would just like to point out that reference n. 65 does not correspond to Aiolfi et al. Authors should check references.

Reviewer 3 Report

The manuscript by Wang et al has been improved.

Minor point: Page 1, line 34

The percentage of people infected with HBV,  at some point in their life,  is estimated to be 30% (or around 2 billion) and not 3%. It is the estimated number of the World's population with serological evidence of current or past HBV infection (Trépo C, Chan HL, Lok A. Hepatitis B virus infection. Lancet. 2014 Dec 6;384(9959):2053-63. doi: 10.1016/S0140-6736(14)60220-8.). The number of chronically infected is significantly lower, as stated, around 250 million.

The language in revised paragraphs still needs attention.
